# Evaluation of the impact of the NICE head injury guidelines on inpatient mortality from traumatic brain injury: an interrupted time series analysis

Carl Marincowitz,[1] Fiona Lecky,[2] Victoria Allgar,[3] Trevor Sheldon[4]

[1]Hull York Medical School, University of Hull, Hull, UK
[2]School of Health and Related Research, University of Sheffield, Sheffield, UK
[3]HYMS/Health Sciences, York University, York, UK
[4]Health Sciences, University of York, York, UK

**Correspondence to**
Dr. Carl Marincowitz;
Carl.Marincowitz@hyms.ac.uk

## ABSTRACT

**Objective** To evaluate the impact of National Institute for Health and Care Excellence (NICE) head injury guidelines on deaths and hospital admissions caused by traumatic brain injury (TBI).

**Setting** All hospitals in England between 1998 and 2017.

**Participants** Patients admitted to hospital or who died up to 30 days following hospital admission with International Classification of Diseases (ICD) coding indicating the reason for admission or death was TBI.

**Intervention** An interrupted time series analysis was conducted with intervention points when each of the three guidelines was introduced. Analysis was stratified by guideline recommendation specific age groups (0–15, 16–64 and 65+).

**Outcome measures** The monthly population mortality and admission rates for TBI.

**Study design** An interrupted time series analysis using complete Office of National Statistics cause of death data linked to hospital episode statistics for inpatient admissions in England.

**Results** The monthly TBI mortality and admission rates in the 65+ age group increased from 0.5 to 1.5 and 10 to 30 per 100 000 population, respectively. The increasing mortality rate was unaffected by the introduction of any of the guidelines. The introduction of the second NICE head injury guideline was associated with a significant reduction in the monthly TBI mortality rate in the 16–64 age group (−0.005; 95% CI: −0.002 to −0.007). In the 0–15 age group the TBI mortality rate fell from around 0.05 to 0.01 per 100 000 population and this trend was unaffected by any guideline.

**Conclusion** The introduction of NICE head injury guidelines was associated with a reduced admitted TBI mortality rate after specialist care was recommended for severe TBI. The improvement was solely observed in patients aged 16–64 years. The cause of the observed increased admission and mortality rates in those 65+ and potential treatments for TBI in this age group require further investigation.

## BACKGROUND

There are approximately 2.5 million cases of traumatic brain injury (TBI) (injury to the brain/functional impairment due to external force) annually in the European Union and

### Strengths and limitations of this study

► This study is the first to use complete national data and the robust quasi-experimental method of interrupted time series analysis to evaluate the impact of the National Institute for Health and Care Excellence head injury guidelines.

► We adjusted our analysis for seasonality, autocorrelation and demographic changes using standard statistical techniques.

► Inpatient mortality was assessed at a population level as national data on emergency department attendance for traumatic brain injury (TBI) was unavailable and the guidelines acted to change the admission threshold for TBI identified by CT imaging.

TBI is a leading cause of death and disability.[1] In higher income countries the epidemiology of TBI has changed from a condition predominantly of younger males resulting from high energy trauma, to older people caused by falls.[2]

One of the important health service challenges is identifying the small proportion of patients with life-threatening TBI among the large number of patients who attend emergency departments (EDs) following head injury (blunt trauma to the head) and then ensure they receive specialist care, including neurosurgery, within a time critical period.[3] Previous research demonstrated correctly configured emergency healthcare systems are required to deliver optimal outcomes for patients with severe TBI.[1 4]

In England, since 2003, three National Institute for Health and Care Excellence (NICE) head injury guidelines have been introduced in order to improve the ED identification and subsequent management of TBI (online supplementary material 1).[3 5–7] These would be expected to reduce TBI deaths and unnecessary hospital admissions. All three guidelines advocated increased CT imaging of head

injured patients that present with a minimally impaired conscious level equivalent to a Glasgow Coma Scale of 13–15. Increased costs from imaging were intended to be offset through reduced hospital admissions.[8] The 2007 guideline additionally recommended that patients with severe TBI should be managed in specialist neuroscience centres. At the time of implementation, concerns were raised that guideline recommendations were based on studies in subgroups and lacked supporting level 1 evidence.[4 9 10] Evaluation of the impact of these guidelines on national rates of TBI admissions and patient outcomes is therefore needed.

We describe the first study to use complete national data and interrupted time series analysis to evaluate the impact of early TBI management guidelines on patient outcomes and admission rates for all severities of TBI.

## METHODS
### Data set
Hospital episode statistics (HES) are collected on all inpatients in England. The Office for National Statistics

(ONS) has computerised International Classification of Diseases (ICD) coding of cause of death information recorded on death certificates.

We used individual patient level HES data provided by NHS Digital on all emergency inpatient hospital admissions in England from April 1998 to April 2017. Reason for admission is recorded using ICD10 coding. For patients with ICD10 diagnostic codes: S00–S09 (indicating TBI) or T04.0 and T06.0 (crushing injury to the head) who died up to 30 days from discharge ONS cause of death was also provided.[11] ONS coding changed from ICD9 to ICD10 in 2001.

### Deaths attributable to TBI
Online supplementary material 2 summarises how deaths attributable to TBI over the study period were identified. A total of 852 646 deaths linked to admissions for head injury were identified by NHS Digital. We searched all cause of death fields for ICD9 and ICD10 codes defined by the Centers for Disease Control and Prevention (CDC) as indicating a death attributable to TBI (table 1).[12] When any of these ICD codes were present the death was coded

**Table 1** Annual numbers of deaths and admissions from traumatic brain injury in England (estimated from data set provided by NHS Digital)

| Year | Admissions all age groups | Admissions 0–15 | Admissions 16–64 | Admissions 65+ | Death all age groups | Deaths 0–15 | Deaths 16–64 | Deaths 65+ |
|---|---|---|---|---|---|---|---|---|
| *1998 | 47 820 | 17 739 | 22 348 | 7631 | 677 | 45 | 307 | 331 |
| 1999 | 63 599 | 23 848 | 29 088 | 10 553 | 964 | 71 | 446 | 453 |
| 2000 | 60 001 | 21 774 | 27 793 | 10 280 | 1076 | 69 | 492 | 525 |
| 2001 | 58 497 | 21 065 | 26 553 | 10 774 | 1105 | 62 | 519 | 532 |
| 2002 | 55 941 | 19 579 | 25 808 | 10 424 | 1178 | 46 | 508 | 634 |
| 2003 | 60 336 | 19 630 | 28 405 | 12 239 | 1294 | 51 | 521 | 729 |
| 2004 | 68 662 | 20 361 | 33 298 | 14 937 | 1342 | 49 | 568 | 734 |
| 2005 | 75 391 | 20 417 | 36 832 | 18 093 | 1484 | 43 | 606 | 840 |
| 2006 | 77 333 | 19 696 | 38 005 | 19 566 | 1570 | 49 | 610 | 917 |
| 2007 | 75 219 | 18 128 | 36 473 | 20 566 | 1665 | 39 | 624 | 1012 |
| 2008 | 74 158 | 17 481 | 34 657 | 21 938 | 1621 | 26 | 564 | 1036 |
| 2009 | 81 218 | 18 111 | 37 178 | 25 848 | 1739 | 35 | 603 | 1105 |
| 2010 | 81 032 | 18 008 | 35 064 | 27 856 | 1817 | 29 | 530 | 1260 |
| 2011 | 82 093 | 18 604 | 33 989 | 29 390 | 1879 | 35 | 500 | 1354 |
| 2012 | 76 925 | 16 453 | 30 475 | 29 901 | 2025 | 27 | 525 | 1474 |
| 2013 | 76 429 | 15 966 | 28 983 | 31 379 | 2204 | 27 | 497 | 1687 |
| 2014 | 79 372 | 15 535 | 28 833 | 34 890 | 2361 | 15 | 462 | 1886 |
| 2015 | 76 648 | 13 630 | 27 517 | 35 357 | 2610 | 18 | 493 | 2102 |
| 2016 | 74 242 | 13 120 | 25 228 | 35 488 | 2682 | 30 | 511 | 2145 |
| *2017 | 16 247 | 2619 | 5483 | 8037 | 504 | | 79 | 420 |

*Data are from April 1998 to March 2017, so 1998 and 2017 are part years and small number have been suppressed in accordance with NHS digital guidance.
ICD9 definition TBI: 800, 801, 803, 804, 850, 851, 852, 853, 854, 905.0, 907.0 and 873 ICD10 definition TBI: S01.0–S01.9, S02.0, S02.1, S02.3, S02.7-S02.9, S04.0, S06.0–S06.9, S07.0, S07.1, S07.8, S07.9, S09.7–S09.9, T01.0, T02.0, T04.0, T06.0, T90.1, T90.2, T90.4, T90.5, T90.8 and T90.

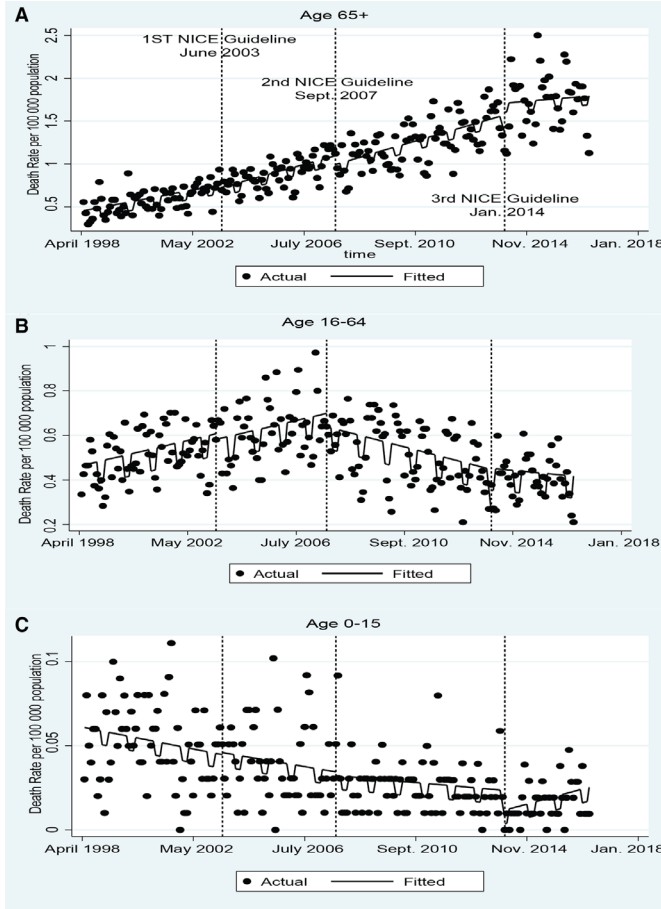

**Figure 1** The impact of the National Institute for Health and Care Excellence head injury guidelines on monthly traumatic brain injury mortality rate per 100 000 population.

as attributable to TBI. A total of 34 659 deaths attributable to TBI were identified, and these were linked to their last recorded admission date as a proxy for when the injury and death occurred. This was not possible for 2862 patients. Neonatal deaths were excluded from analysis due to differences in cause of death coding.

## Admissions attributable to TBI

The same ICD10 codes were used to identify patients admitted with TBI (table 1).[12] We searched the primary diagnostic field in the inpatient HES data set for these codes and when present the reason for admission was coded as due to TBI. Data were cleaned and continuous inpatient spells (CIPS) were created for patients admitted with TBI using the approach outlined by Castelli, Laudicella and Street as this includes transfers within CIPS.[13] 1361537 CIPS for TBI were identified for 1245720 patients. Following cleaning, 402 CIPs were found to have admission dates prior to April 1998 and were excluded. Demographic and comorbidity information was calculated from the first consultant episode of a CIP. This included the monthly proportion of TBI admissions for males, monthly median age of admissions and mean monthly admission Charlson comorbidity index score (using ICD10 code definitions and weights

used to calculate the summary hospital-level mortality indicator).[14] This was compared with adjustment using a modified Charlson comorbidity index derived from the national (Trauma Audit and Research Network; TARN) trauma registry.[15]

## Outcomes

The monthly number of patients with deaths and admissions attributable to TBI between April 1998 and March 2017 was calculated. These were stratified into guideline specific age groups: 0–15, 16–64 and 65+. Monthly mortality and admission rates were calculated per 100 000 population using Nomis ONS mid-year population estimates for England for each age group.[16]

## Statistical analysis

A monthly time series of the mortality rate for TBI was plotted for the study period. Interrupted times series (ITS) analysis was conducted assessing the impact of the NICE guidelines using the ITSA package in STATA V.14.[17] ITS analysis is a robust and increasingly used quasi-experimental method for the evaluation of health policies and allows causality to be attributed to an intervention introduced at a specific time point.[18]

The ITS model included three intervention time points corresponding to the introduction of each guidelines in: June 2003, September 2007 and January 2014. Analysis was conducted separately for the 0–15, 16–64 and 65+ age groups. A segmented regression model predicting the mortality rate and hospital admission rate for TBI per 100 000 population in each age group per month was estimated.[18] A discontinuity in the gradient (trend) or intercept (level) of the fitted model was tested for at the time point when each guideline was introduced, and discontinuities in the model were measured in the monthly rate of the outcome per 100 000 population.

To adjust for potential changes in the composition of the TBI population that could possibly affect the risk of mortality a further ITS model predicting the TBI mortality rate adjusted for % male, median age and mean Charlson comorbidity index score of patients admitted with TBI was fitted. Stratification by age group and intervention points were identical to the previous analysis.

In all analyses, autocorrelation of the residuals was assessed using the Durbin-Watson and Rho statistic. Throughout we used the Prais-Winsten transformation adjustment for auto-correlation due to improved fit of the model, deviation from a Durbin-Watson statistic of 2 and a non-statistically significant Rho statistic.[18] Seasonality was assessed by introducing a dummy variable to the model in which winter months (December, January and February) were coded 1 and was included in the model when statistically significant.[19] To assess for possible implementation lags a sensitivity analysis was performed for all models in which the 12 months immediately following the introduction of a guideline were removed.[18]

**Table 2** The impact of the National Institute for Health and Care Excellence head injury guidelines on monthly traumatic brain injury mortality rate per 100 000 population

| Age band | Winter effect | Initial trend | First NICE guideline | Second NICE guideline | Third NICE guideline | Durbin-Watson statistic |
|---|---|---|---|---|---|---|
| 65+ | −0.1 (95% CI:−0.16 to −0.04) p<0.01 | 0.005 (95% CI:0.002 to 0.008) p<0.01 | Change level: −0.034 (95% CI:−0.21 to 0.14) p=0.71 Change trend: 0.002 (95% CI:−0.003 to 0.008) p=0.43 | Change level: −0.1 (95% CI: −0.27 to 0.07) p=0.24 Change trend: 0.0004 (95% CI: −0.005 to 0.006) p=0.89 | Change level: 0.13 (95% CI:−0.04 to 0.32) p=0.14 Change trend: −0.005 (95% CI:−0.01 to 0.002) p=0.14 | Untransformed 1.57 Prais-Winsten 1.86 |
| 16–64 | −0.1 (95% CI: −0.13 to −0.06) p<0.01 | 0.002 (95% CI:0.001 to 0.004) p<0.01 | Change level: −0.03 (95% CI: −0.11 to 0.06) p=0.57 Change trend: −0.00002 (95% CI: −0.003 to 0.003) p=0.99 | Change level: −0.06 (95% CI: −0.15 to 0.003) p=0.17 Change trend: −0.005 (95% CI:−0.007 to −0.002) p<0.01 | Change level: 0.005 (95% CI:−0.087 to 0.096) p=0.92 Change trend: 0.002 (95% CI:−0.002 to 0.005) p=0.38 | Untransformed 1.79 Prais-Winsten 1.95 |
| 0–15 | −0.01 (95% CI:−0.01 to −0.003) p<0.01 | −0.0003 (95% CI: −0.0005 to −0.00001) p=0.04 | Change level: 0.001 (95% CI: −0.01 to 0.01) p=0.18 Change trend: 0.00004 (95% CI:−0.0004 to 0.0004) p=0.17 | Change level: −0.0021 (95% CI: −0.01 to 0.01) p=0.74 Change trend 0.0001 (95% CI:−0.0003 to 0.0005) p=0.58 | Change level: −0.01 (95% CI:−0.03 to 0.002) p=0.09 Change trend: 0.0005 (95% CI: −0.00005 to 0.001) p=0.08 | Untransformed 2.12 Prais-Winsten 1.99 |

## Patient and public involvement

The Hull and East Yorkshire NHS Trust Trans-Humber Consumer Research Panel and Hull branch of the Headway charity were consulted in the initial stages of developing the research questions addressed in this study. These patient groups highlighted that although national head injury guidelines seemed evidence based, there appeared to be little evidence to show they had achieved their aims.

## RESULTS
### Mortality rate

Table 1 shows the annual number and online supplementary material 3 shows the annual rates of deaths and hospital admissions for TBI. The proportion of all TBI annual admissions for patients 65+ increased from 17% in 1998 to 48% in 2016 and the proportion of all TBI deaths in this age group increased from 49% to 78% over the same period. Figure 1 shows the monthly mortality rate per 100 000 population in each age group. Table 2 shows the results of the unadjusted interrupted time series assessing the impact of the NICE head injury guidelines. Deaths were more likely to occur in non-winter months in all age groups and so the figures are seasonally adjusted.

The trends in mortality rate and impact of the guidelines varied between age groups. In the 65+ age group the monthly TBI mortality rate increased from around 0.5 to over 1.5 per 100 000 population over the time period (figure 1A). This was accompanied by an increase in the Charlson score of patients 65+ admitted with TBI (online supplementary material 4). The NICE head injury guidelines were not associated with statistically significant changes in the level or trend in the mortality

rate (table 2). Subgroup analysis of patients aged 65–84 and 85+ showed that the increase in the mortality rate was greater in those 85+, from around 1 to over 6 per 100 000 population but similar changes were associated with the introduction of the guidelines to the whole 65+ population (online supplementary material 5).

The second guideline was found to be associated with a large reduction in mortality in the 16–64 age group (figure 1B). Before the guideline, the monthly mortality rate was increasing but the introduction of the second NICE guideline is associated with a reversal of this trend (−0.005; 95% CI:−0.002 to −0.007) (table 2). The reduction in mortality appears to slow at the time of the introduction of the third NICE guideline but this was not statistically significant. There was an increase in age of patients in the 16–64 age group admitted with TBI but no change in the Charlson comorbidity score over the period (online supplementary material 4).

In the 0–15 age group the mortality rate fell continuously over the time period from around 0.05 to 0.01 per 100 000 population (figure 1C). There were fewer monthly numbers of deaths and so more random variability in rates. None of the guidelines were associated with a statistically significant change in the level or trend in the mortality rate (table 2), though the high random variability meant we had lower statistical power to detect such changes as statistically significant.

Adjustment for the monthly median age, mean Charlson Score and proportion of male admissions for TBI did not materially alter the estimates associated with the introduction of guidelines in any of the age groups (online supplementary material 6). In the 16–64 age group the estimate of the reversal in trend in mortality rate associated with

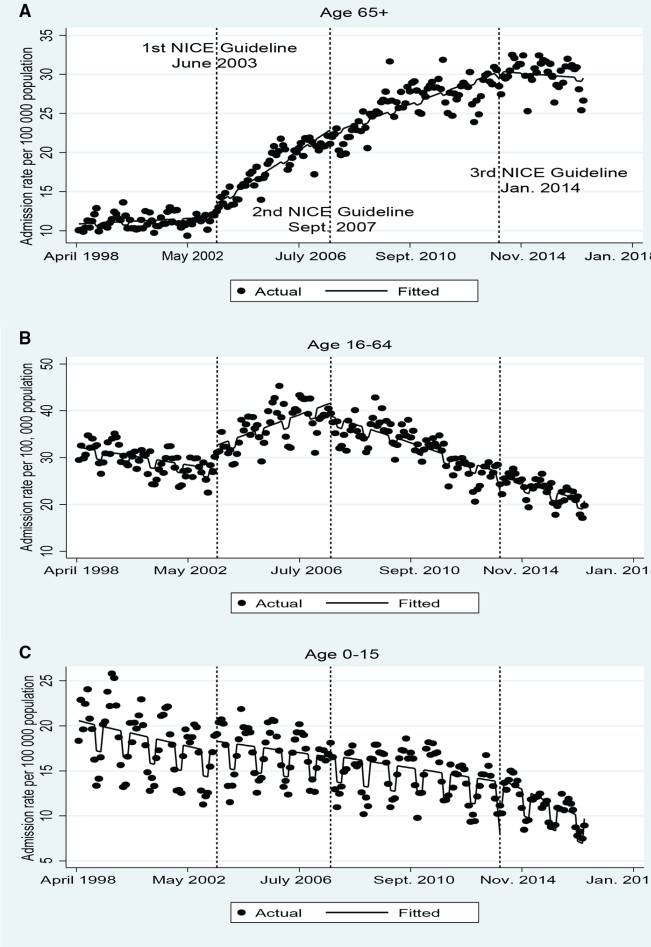

**Figure 2** The impact of the National Institute for Health and Care Excellence head injury guidelines on monthly traumatic brain injury hospital admissions per 100 000 population.

the second guideline, −0.006 (95% CI:−0.008 to −0.003), was similar to the unadjusted analysis. The levelling off in the rate of reduction in mortality in the 16–64 age group associated with the third NICE guideline became marginally statistically significant, although the estimate is similar, 0.003 (95% CI: 0.00005 to 0.007). No adjustment was made for the standard Charlson score in the paediatric and 16–64 age groups as it did not change over time. The monthly mean trauma modified Charlson score in the 16–64 age group increased slightly from 0 to 1 and adjustment for this increased the estimated size of reversal in mortality trend associated with the second NICE guideline, −0.008 (95% CI: −0.01 to −0.005), (online supplementary material 4). The sensitivity analysis for the effect of implementation lags did not affect the estimates associated with the introduction of any guideline (online supplementary material 7).

### Admission rate

Figure 2 shows the trends in monthly TBI admissions stratified by age group and table 3 presents estimates of the change in admission rate associated with the introduction of each head injury guideline iteration. The admission rate increased threefold (from around 10 per 100 000 to 30 per 100 000) in the 65+ age group. The introduction of the first NICE guideline is associated with large increasing trends in monthly TBI admissions per 100 000 population in both the 65+ age group (0.17: 95% CI: 0.11 to 0.22) and the 16–64 age group (0.25: 95% CI: 0.16 to 0.34) (table 3).[20] The subsequent two guidelines are associated with significant reductions in this trend and admission rates level off following the third guideline in the 65+ age group (table 3 and figure 2A). In the 16–64 age group, the TBI admissions trend reverses and declines

**Table 3** The impact of the National Institute for Health and Care Excellence head injury guidelines on monthly traumatic brain injury hospital admission rate per 100 000 population

| Age band | Winter effect | Initial trend | First NICE guideline | Second NICE guideline | Third NICE guideline | Durbin-Watson statistic |
|---|---|---|---|---|---|---|
| 65+ | −0.44 (95% CI: −0.94 to 0.06) p=0.08 | 0.01 (95% CI: −0.02 to 0.05) p=0.42 | Change level: 1.71 (95% CI:−0.01 to 3.44) p=0.05 Change trend: 0.17 (95% CI: 0.11 to 0.23) p<0.01 | Change level: −0.4 (95% CI: −2.08 to 1.27) p=0.64 Change trend: −0.08 (95% CI: −0.13 to −0.03) p<0.01 | Change level: 0.04 (95% CI:−1.73 to 1.82) p=0.96 Change trend: −0.13 (95% CI:−0.2 to −0.05) p<0.01 | Untransformed 1.1 Prais-Winsten 2.09 |
| 16–64 | −1.92 (95% CI: −2.77 to −1.07) p<0.01 | −0.08 (95% CI: −0.13 to −0.02) p<0.01 | Change level: 5.21 (95% CI: 2.53 to 7.89) p<0.01 Change trend: 0.25 (95% CI: 0.16 to 0.34) p<0.01 | Change level: −2.76 (95% CI:−5.35 to −0.16) p=0.04 Change trend: −0.33 (95% CI: −0.42 to −0.25) p<0.01 | Change level: −0.72 (95% CI: −3.49 to 2.03) p=0.61 Change trend: 0.02 (95% CI:−0.09 to 0.13) p=0.73 | Untransformed 1.35 Prais-Winsten 2.11 |
| 0–15 | −2.87 (95% CI: −3.40 to −2.34) p<0.01 | −0.06 (95% CI:−0.11 to −0.01) p=0.03 | Change level: 1.3 (95% CI: −1.03 to 3.63) p=0.27 Change trend: 0.02 (95% CI: −0.07 to 0.11) p=0.61 | Change level: 0.19 (95% CI: −2.09 to 2.47) p=0.87 Change trend −0.005 (95% CI: −0.08 to 0.08) p=0.91 | Change level: 0.34 (95% CI:−2.03 to 2.72) p=0.78 Change trend: −0.08 (95% CI:−0.19 to 0.03) p=0.17 | Untransformed 1.07 Prais-Winsten 1.70 |

after the second NICE guideline (−0.33: 95% CI: −0.42 to −0.25) (table 3 and figure 2B).

In the 0–15 age group TBI admissions steadily fall over the study period from around 20 per 100 000 to 10 per 100 000 (figure 2C), and is unaffected by the introduction of the guidelines (table 3).

A sensitivity analysis for implementation lags in which the 12 months following the introduction of a guideline were removed from the analysis did not materially change the estimates associated with the introduction of the guidelines in any age group (online supplementary material 8).

## DISCUSSION

### Summary

To our knowledge this is the first study to use national population based data and interrupted time series analysis to evaluate the impact of the NICE head injury guidelines in England. The second NICE guideline was associated with a reduction in the admitted TBI mortality rate in the 16–64 age group at a population level (table 2). We found no other impact on mortality associated with the three guideline iterations.

There was a continual and significant increase in TBI mortality and admission rates in the 65+ age group and a contrasting falling trend in mortality and admission rates in children (figure 1 and figure 2). Both trends began before the introduction of the NICE guidelines and were not significantly affected by any of the three iterations. In both the 16–64 and 65+ age groups there was a large increase in hospital admissions for TBI at the time the first NICE guideline was introduced (figure 2).

Increased imaging was intended to reduce hospital admissions by reducing diagnostic uncertainty but the first NICE guideline coincided with the introduction of the 4 hour target.[8 20] We have shown, using Scottish data assessing the impact of similar Scottish Intercollegiate Guidelines Network (SIGN) guidelines (introduced at a different time to the 4-hour target), that the 4-hour target acted to undermine this reduction and cause a large increase in hospital admissions.[21] No mortality benefit was found at the time of the introduction of the 4-hour target in England.

Later guidelines were associated with a reduction in hospital admissions rates in both adult populations assessed (figure 2). Further increases in CT imaging may have reduced hospital admissions, as intended, by reducing diagnostic uncertainty in the ED, without the distorting effect of the 4-hour target introduction.

### Strengths

We used complete national data for England to assess the impact of the NICE head injury guidelines on mortality after admission for TBI at a population level. We have used individual level patient data to define TBI deaths and admissions. We controlled for seasonal factors and auto-correlation using established techniques.[18] We used mid-year population estimates to adjust for changes in the demography of England's population.

### Weaknesses

Ideally, we would have estimated the impact of the guidelines on case fatality, as this better measures the impact on the population at risk. The impact on case fatality of those attending ED with TBI could not be estimated because ED data were not collected until 2007. The impact on case fatality of those admitted with TBI could be estimated but because the guidelines resulted in changes in admissions policies and rates, the rate of deaths per admission is difficult to interpret. Instead we analysed the impact on the population TBI mortality rate, as this represents the best available unbiased measure of the guidelines' impact. This outcome may be affected by changes in the underlying population TBI rate that we are unable to account for, although annual attendances to the ED for head injury gradually smoothly increased over the study period (online supplementary material 9). We were unable to assess possible impact on disability or other patient reported outcomes, as they are not routinely collected.

ONS linked HES data is based on routinely collected administrative data; these can suffer from poor accuracy of injury coding.[22] This is particularly likely in older patients with multimorbidity (TARN, personal communication 2018). Random poor coding, as opposed to a discrete and systematic change in coding practice, however, is unlikely to account for discontinuities observed at the specific time points of interest but may make a discontinuity harder to detect. ONS changed from ICD9 to ICD10 coding of cause of death in 2001. A sensitivity analysis excluding the period that used ICD9 coding did not materially alter the estimate of the reversal in mortality trend associated with the second guideline in the 16–64 age group. We are unaware of other significant changes to coding practice in the HES or ONS data during the study period. The limitations of HES data mean that mortality rates could not be adjusted for anatomical severity of brain injury and presenting physiology. However, adjustment for other known predictors of TBI mortality did not materially change estimates associated with the introduction of the guidelines and we are unaware of evidence that the prevalence of these factors changed at the point individual guidelines were introduced.

The impact of guidelines is limited by how well they are implemented. The NICE head injury guidelines have been found to be well implemented,[23] although with less compliance to CT imaging recommendations in the paediatric population.[23 24] There is evidence that each guideline caused step increases in CT head scanning in other age groups, particularly in those 65+.[10 25]

The reconfiguration of the trauma network in England in 2012 is a co-intervention which could affect the TBI mortality rate.[26] However, we found no impact on mortality associated with the 2014 NICE guideline introduced around this time. Apart from the introduction of the 4-hour ED admissions target in 2004, we are unaware

**Table 4** Comparison to previous literature

| Previous study | | | Current study |
|---|---|---|---|
| | **Study population** | **Findings** | **Findings** |
| Fuller *et al*, 2009[4] | TARN eligible patients at TARN submitting hospitals (approx. 50% England) between 2003 and 2009. | From the period 2004 onwards as the proportion of patients with TBI transferred and managed in neuroscience centres increased and the risk adjusted mortality rate for TBI fell. | Complete national data for all hospital in England. A reversal in trend in the mortality rate in the 16–64 age group when the second NICE guideline recommending management of patients with severe injuries in specialist centres was introduced. |
| Marlow *et al*[24] | Patients aged <16 with ICD10 codes indicating head injury admitted to hospitals in England between 2000 and 2011. | Assessed the annual rate of inpatient deaths (all-cause mortality) for patients admitted with ICD10 codes indicating head injury. Found the death rate fell across the time period, but there was only a statistically significant reduction in the death rate after the 2007 NICE head injury guideline. | The inpatient TBI mortality rate (as indicated by coding of death certificates) for patients aged <16 fell from 1998 to 2017 and was unaffected by the introduction of the NICE guidelines. |
| The Trauma Audit and Research network report: major trauma in older people[25] | TARN eligible patients at TARN submitting hospitals between 2005 and 2014 (all hospitals in England by 2014) | A large increase in major trauma, including TBI, in patients 65+, disproportionate to UK population demographic changes. Hypothesised due to increased case ascertainment due to more liberal CT imaging. | We found a large increase in the admission rate for TBI in those 65+ from 10 per 100 000 population to 30 per 100 000 population between 2002 and the point the third NICE guideline was introduced in 2014. |

of any other co-interventions that occurred around the time the NICE guidelines were introduced which could account for the observed discontinuities in mortality and hospital admissions.

### Comparison to previous literature

Few previous studies assess the impact of the NICE head injury guidelines (see table 4).[9] A cohort study using TARN national registry data suggested the increased rate of transfer of severe TBI patients to neuroscience centres between 2003 and 2009 was associated with a halving of severe TBI case fatality.[4] TARN data were only collected at approximately half of hospitals in England until 2012 and on a TBI patient subset. Our study, using complete national data and interrupted time series analysis, found that guideline recommended management of patients with severe injuries in specialist centres only reduced the mortality rate in the 16–64 age group.

A paediatric study analysing English HES data from 2000 to 2011 found a reduction in annual mortality during admissions for head injury after the introduction of 2007 NICE guideline.[24] We found a fall in the mortality rate over the study period in the 0–16 age group which was unaffected by any guideline. This may reflect the greater number of data points we used to estimate the time-dependent model and use of interrupted time series

analysis to assess for discontinuities. We also used ONS linked HES data to identify deaths directly attributable to TBI up to 30 days following discharge. The observed decreasing mortality and admission rates may reflect improving clinical management or a reduction in TBI in this age group due to improving road traffic safety during the study period.[24]

An economic evaluation of the NICE guidelines found them to be cost effective due to a reduction in hospital admissions predicted from early single centre studies and improved outcomes.[8 10] A subsequent study using HES data found hospital admissions for head injury increased after the introduction of the first NICE guideline.[11] The similar increase in adult TBI admissions we found associated with the first NICE guideline probably is due the 4-hour target.[21] We found subsequent NICE guidelines improved outcomes and reduced hospital admissions in the 16–64 but not the 65+ age group, implying the guidelines were less cost effective in older patients.

Other studies using TARN data have found increases in TBI in patients 65+ disproportionate to population changes and it has been suggested that better case ascertainment due to increased CT imaging in older patients may account for this.[2 25] The large increase in admissions for TBI for those 65+ we found at the point the first

guideline was introduced, although boosted by the 4-hour target, supports this (figure 2A and table 3). The lack of improvement in admitted TBI mortality in older patients following the second NICE guideline could either result from unequal access to treatment in specialist centres or such treatment appearing to be less effective in this group. The TARN older persons audit found patients aged over 60 to be less likely to be managed in major trauma centres (where neurosurgical units are located in England) and more likely to experience delays in investigation and be treated by junior staff.[25] However, other studies have found age to be an independent predictor of mortality that is unaffected by early treatment in neuroscience centres.[27 28]

We are unaware of comparable national evaluations of the impact of head injury guidelines. Evaluations of International Brain Trauma Foundation guidelines, particularly in the USA, have utilised evidence from single centre studies or subsets of patients.[23 29 30] Evaluation of their national impact has not been possible due to their variable implementation.[23 30]

## Implications

We found evidence that only the second NICE head injury guideline was associated with a change in population-based TBI mortality. This guideline contained a recommendation for increased management of severe TBI in specialist centres. Much research has focused on determining which head injured patients require CT imaging.[3 31] Increased diagnosis by itself, however, without a change in subsequent patient management was not associated with improved outcomes in our analysis. Even if apparent increases in TBI rates in older patients reflect the identification of previously unmet need, this still represents a significant health service challenge. Routine ICD coding of TBI is particularly problematic in this group and robust evaluation of treatment in specialist neuroscience centres and other interventions may be required to improve outcomes in older TBI patients. The UK, however, has one of the lowest numbers of ICU beds per population in Europe and when the 2007 guideline recommendation was made concerns were raised about the system meeting demand.[9 32] Research needs to focus on how to best configure and ration specialist services for TBI in a transparent and evidence-based way.

## CONCLUSION

This first national evaluation suggests that the introduction of the second NICE head injury guideline was associated with a reduction in the admitted TBI mortality rate in the 16–64 age group and a reduction in TBI admissions in England. The guidelines were not associated with significant changes in the secular trend for TBI admissions and subsequent mortality in children and those aged 65+. Research is needed to identify clinically and cost-effective management approaches for TBI in older patients.

**Acknowledgements** The Hull and East Yorkshire NHS Trust Trans-Humber Consumer Research Panel and Hull branch of the Headway charity helped develop the research questions addressed in this study.

**Contributors** This idea for the study was conceived by CM with help from TS, FL and VA. The analysis was completed by CM with specialist advice regarding interrupted time series analysis from TS and VA. FL provided specialist advice regarding the clinical context and interpretation of the results. All authors read and approved the final manuscript.

**Funding** CM is funded by a National Institute for Health Research Doctoral Fellowship (DRF-2016-09-086). This study presents independent research funded by the National Institute for Health Research (NIHR). The views expressed are those of the author(s) and not necessarily those of the NHS, the NIHR or the Department of Health and Social Care.

**Competing interests** None declared.

**Patient consent for publication** Not required.

**Ethics approval** This study involved the analysis of anonymised routinely collected data, and therefore NHS Research Ethics Committee review was not required. Data were stored and processed in accordance with NHS Digital guidance and data sharing agreement.

**Provenance and peer review** Not commissioned; externally peer reviewed.

**Data sharing statement** Access to the individual level Office of National Statistics linked Hospital Episode Statistics is subject to a data-sharing agreement with NHS Digital that limits access to the data to named members of the research team.

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
