## [Reviewer comments · BMJ Open]

ARTICLE DETAILS

TITLE (PROVISIONAL)	An evaluation of the impact of the NICE head injury guidelines on inpatient mortality from traumatic brain injury: an interrupted time series analysis
AUTHORS	Marincowitz, Carl; Lecky, Fiona; Allgar, Victoria; Sheldon, Trevor

VERSION 1 - REVIEW

REVIEWER	Shu-Ling Chong KK Women's and Children's Hospital Singapore
REVIEW RETURNED	25-Jan-2019

GENERAL COMMENTS	Thank you for the opportunity to review your manuscript. The authors perform an interrupted time series analysis to evaluate the impact of three NICE head injury guidelines in England. They used national level data from the Hospital Episode Statistics (HES) and Office for National Statistics (ONS) and detailed their process of data linkage. They recognised the need to adjust for confounders in the segmented regression model predicting for mortality rate, and performed a sensitivity analysis accounting for lag in effect. My main comments would be the following: Besides the introduction of the NICE guidelines, the authors recognise that other interventions may have accounted for changes in mortality and admission rates over such a long period of time, specifically the 4-hour target in England. (The authors also refer to the reconfiguration of the trauma network in England.) What other interventions had taken place during this time? It is apparent from the downward trend of mortality and admission rate in the 0-15 year olds (unaffected by the NICE guidelines) that there are likely other ongoing interventions. This would be important to account for in the analysis. There is an assumption that the effects seen are secondary to the correct implementation of the NICE guidelines - the authors state (Page 14, lines 54-55) that the "NICE head injury guidelines have been found to be well implemented" but cite a single study (ref 28) which looks at predictors of in-hospital mortality and 6-month functional outcomes in older adults. This point needs to be reinforced with better references, since the authors assume that the NICE guidelines can account for the change in mortality and admission rates (and therefore must be closely adhered to, nationally)
--

	Under the Discussion, the comparison to prior literature (especially the TARN data) can be better described using a table comparing and contrasting their findings. Currently it is a little difficult for the reader to follow. Minor comment: Table 1 should be represented in rates instead of annual numbers.
--	---

REVIEWER	Hui Chen Capital Medical University, Beijing China
REVIEW RETURNED	16-Feb-2019

GENERAL COMMENTS	The article “The effect of the NICE head injury guidelines on inpatient mortality from traumatic brain injury: an interrupted time series analysis” used complete national data and interrupted time series analysis to evaluate the impact of the NICE head injury guidelines. This article maybe useful for making decisions on health service and resource allocation. Minor comments  1. It's better to provide some comparisons of mortality and/or admission rate among age groups. 2. The definition of changes in the level in the mortality/admission rate and the statistical method for the comparison were not stated clearly in the manuscript. 3. Some word expressions were not consistent throughout the manuscript. For example, the usage of over 65, 65+, and ≥65, or age group and age grouping.
---

REVIEWER	Jacob Simmering Carver College of Medicine University of Iowa USA
REVIEW RETURNED	20-Feb-2019

GENERAL COMMENTS	Marincowitz et al present an interesting analysis of the effects of 3 rounds of NICE guidelines for the management of TBI on the rate of hospitalization and mortality from TBI. Specifically, they use an interrupted time series approach to model the effects of guideline publication on the incidence series. They found no effect of any of the three guidelines on incidence or mortality for those over 65 years or under 15 years in age. Among those aged 16 to 64, the authors found a decrease in admissions following the introduction of the 2nd guideline as well as a decrease in mortality. The modeling seems to address the unique properties of time-series data. However, I have some concerns about how the outcome is measured. The authors discuss their reasoning for using absolute incidence and mortality per 100,000 population as being interpretable measures and having issues with not being able to assess case fatality rates or the rate of presentation to the ED for TBI. I understand the issues that may be created by using inpatient mortality while inpatient admission probabilities are directly affected by the guidelines. However, I am concerned that the underlying rate of TBI may be extremely variable across the study interval. It seems very plausible that the rate of TBI in the population changed. For instance, between the year 2000 and
--

2008, the number of vehicle miles traveled in the UK increased by about 8% and decreased by 5-6% during the recession following 2008. It seems that vehicle accidents may be a common cause of TBI in adults 16-64 and if vehicle usage changes (or if the safety systems onboard the vehicles changes) that the number of TBI will change.

I believe the authors can address this concern by exploring their ED data. While that data is only present from year 2007 onward, they can use that data to examine whether the underlying rate of TBI is constant during the latter half of the study interval. Showing that TBI rates in the population are roughly constant will make the argument that there are changes in admission probability or mortality easier to believe.

Other comments:

1. Do the causes of the TBI vary between age groups and over time? The authors mention that older adults (65+) tend to represent greater numbers of falls while among younger adults it is predominately men presenting with TBI. Are there external cause of injury codes (e-codes) that can be used to assess the changes in the causes of TBI over time? It seems that the causes are changing as the age and gender mix is variable over time.
2. The methods section regarding the model was confusing. Was a separate model with different intercepts/slopes for each time period as well as adjustments for % male, median age and mean Charlson score fit for each age group? Are these the models for which the results are shown in the tables and figures in the main paper? What was the objective of the monthly time series analysis described in the second paragraph on page 8?
3. In the discussion, the authors detail what changes they'd expect from the guidelines (reduced admission rates following CT imaging, reduced mortality following early diagnosis and transfer to an appropriate level of care). It is would nice to introduce these expected directions of change earlier in the paper.
4. Consider removing the 65+ and 0-15 analysis from this paper. It seems that the treatment methodology for 65+ is likely different from younger people due to greater levels of comorbidity and frailty. They may be less sensitive to changes in guideline care due to already high levels of CT use and transfer. Likewise for the youngest members on the sample. Those 16-64 likely represent a population at greater probability of being affected by changes in guidelines for care of TBI.

Minor Comments/Typos:

1. Page 9, sentence "the proportion of all TBI annual admissions and deaths in patients over 65 increased from 17% and 49% in 1998 to 48% and 78% in 2016" is confusing. I believe you are saying that in 1998, admissions for TBI in those over 65 accounted for 17% of all TBI admissions which increased to 48% in 2016 and of those who died from a TBI, 49% were over 65 in 1998 while 78% were in 2016. Consider revision.
2. Throughout the paper, either a separator in numbers is omitted (e.g., 100000) or both a comma and space (e.g., 100, 000) are used. I'd encourage use of a separator in order to make the numbers more easily understand and using either a space or a comma and not both.

VERSION 1 – AUTHOR RESPONSE

Reviewer 1:

Besides the introduction of the NICE guidelines, the authors recognise that other interventions may have accounted for changes in mortality and admission rates over such a long period of time, specifically the 4-hour target in England. (The authors also refer to the reconfiguration of the trauma network in England.) What other interventions had taken place during this time? It is apparent from the downward trend of mortality and admission rate in the 0-15 year olds (unaffected by the NICE guidelines) that there are likely other ongoing interventions. This would be important to account for in the analysis.

We are unaware of any other health policies or co-interventions which occurred around the discrete time points at which the guidelines were introduced and could account for the discrete changes in trend or level at those time points and associated with the introduction of the guidelines. We now explicitly state this in the last paragraph of the weaknesses section (page 16) . Our aim was to evaluate the impact of the NICE head injury guidelines and not necessarily explain overall trends in TBI mortality and admissions over time.

The downward trend in mortality and admission rate in the 0-15 age group may reflect improving clinical management and a reduction in trauma in this age group associated with improved road traffic safety and other health and safety measures. We have added a sentence at the end of the second paragraph of the section entitled comparison to previous literature (page 17) to highlight this as a potential explanation for the trend observed in the 0-15 age group.

There is an assumption that the effects seen are secondary to the correct implementation of the NICE guidelines - the authors state (Page 14, lines 54-55) that the "NICE head injury guidelines have been found to be well implemented" but cite a single study (ref 28) which looks at predictors of in-hospital mortality and 6-month functional outcomes in older adults. This point needs to be reinforced with better references, since the authors assume that the NICE guidelines can account for the change in mortality and admission rates (and therefore must be closely adhered to, nationally)

Thank you for pointing out our error; we mistakenly cited reference 28 for this statement when we intended to cite reference 23. Reference 23 is a systematic review including 5 studies which assessed adherence to the NICE head injury guidelines and found they are strongly adhered to and are amongst the most implemented head injury guidelines internationally. We cite Hassan et al (reference 10) and the TARN older persons report (reference 25) in the same paragraph and these both present evidence that CT imaging of head injured patients in the ED increased at the time the NICE head injury guidelines were introduced.

Under the Discussion, the comparison to prior literature (especially the TARN data) can be better described using a table comparing and contrasting their findings. Currently it is a little difficult for the reader to follow.

We have added a table comparing and contrasting the findings as requested - Table 4

Minor comment: re

Table 1 should be represented in rates instead of annual numbers.

The interrupted time series analysis uses monthly rates derived from annual numbers of TBI related hospital admissions and deaths. Presenting the annual numbers in Table 1 allows other researchers to use the reported data to estimate rates and other possible derived outcome measures.

As requested, we have now also presented the annual admission and death rate for TBI stratified by age group in the Supplementary Material 3.

Reviewer 2

Minor comments

1. It's better to provide some comparisons of mortality and/or admission rate among age groups.

We have added supplementary material 3 which presents the annual mortality and admission rate in each age group.

2. The definition of changes in the level in the mortality/admission rate and the statistical method for the comparison were not stated clearly in the manuscript.

We have added the 3rd sentence of the paragraph entitled statistical analysis (page 9) to define changes in level and trend and how we assessed them.

3. Some word expressions were not consistent throughout the manuscript. For example, the usage of over 65, 65+, and ≥ 65 , or age group and age grouping.

We have revised the manuscript throughout to ensure the consistent use of 65+ and age group.

Reviewer 3

However, I have some concerns about how the outcome is measured. The authors discuss their reasoning for using absolute incidence and mortality per 100,000 population as being interpretable measures and having issues with not being able to assess case fatality rates or the rate of presentation to the ED for TBI. I understand the issues that may be created by using inpatient mortality while inpatient admission probabilities are directly affected by the guidelines. However, I am concerned that the underlying rate of TBI may be extremely variable across the study interval. It seems very plausible that the rate of TBI in the population changed. For instance, between the year 2000 and 2008, the number of vehicle miles traveled in the UK increased by about 8% and decreased by 5-6% during the recession following 2008. It seems that vehicle accidents may be a common cause of TBI in adults 16-64 and if vehicle usage changes (or if the safety systems onboard the vehicles changes) that the number of TBI will change.

I believe the authors can address this concern by exploring their ED data. While that data is only present from year 2007 onward, they can use that data to examine whether the underlying rate of TBI is constant during the latter half of the study interval. Showing that TBI rates in the population are roughly constant will make the argument that there are changes in admission probability or mortality easier to believe.

Thank you for your useful comments; we believe you highlight a possible weakness of our study design in that variability in the TBI rate may affect our estimates of absolute mortality. However, we think that this is unlikely to significantly weaken our interpretation of the ITS analysis for two reasons.

As suggested, we have examined the ED data available to us in the NHS Digital annual reports. These present the annual number of patients presenting to the ED with a primary diagnosis of head injury (TBI is not coded) and we summarise the findings in the table in the file attached with this response and Supplementary Material 9. It is important to note that until 2012 not all hospitals submitted ED data (around 25% of expected ED data was not submitted in 2007) and the increasing

number of recorded attendances until 2012 reflects an increase in the number of hospitals submitting data.

In period before 2012 the proportion of ED attendances accounted for by head injury increased slowly over time. There is no reduction in the proportion of head injury attendances associated with the recession in 2008. After 2012, when data were collected at all hospitals in England, both the absolute number and proportion of ED attendances for head injury remain relatively constant. Therefore, in the ED data available to us, there is no evidence that the incidence of head injury and therefore TBI was extremely variable during our study period. The exception to this is 2014 when there is a dip in annual ED attendance for head injury. We found no significant discontinuity when assessing the impact of the 2014 NICE guideline. Apart from discontinuities associated with the introduction of NICE guidelines we have found quite smooth trends in TBI mortality and admission rates. A high degree of variability in the underlying incidence of TBI would be expected to lead to a high degree of variability in admissions and inpatient mortality resulting from TBI across the whole study period.

Second, even if we had found a high level of variability in the population TBI rate, though this may have affected our estimated mortality and admission rate, it would not explain the sharp discontinuity (or reversal) in mortality trend associated with the introduction in 2007 of the 2nd NICE head injury guideline in the 16-64 age group. This is prior to the effects of the recession.

We have added the penultimate sentence of the first paragraph of the weaknesses section (page 16) and Supplementary Material 9 to highlight this potential weakness.

Other comments:

1. Do the causes of the TBI vary between age groups and over time? The authors mention that older adults (65+) tend to represent greater numbers of falls while among younger adults it is predominately men presenting with TBI. Are there external cause of injury codes (e-codes) that can be used to assess the changes in the causes of TBI over time? It seems that the causes are changing as the age and gender mix is variable over time.

There is good evidence that the mechanism of injury in TBI varies between age groups and we have cited studies which show this (Lawrence et al 2016 and the TARN older persons Report). It not possible for us to assess the mechanism of injury in the data available to us. However, as shown in Supplementary Material 2, there were no step changes in age or gender (and so likely mechanism of injury) at the point at which the NICE guidelines were introduced which could account for the discontinuity in trends in outcomes we have shown associated with guidelines.

2. The methods section regarding the model was confusing. Was a separate model with different intercepts/slopes for each time period as well as adjustments for % male, median age and mean Charlson score fit for each age group? Are these the models for which the results are shown in the tables and figures in the main paper? What was the objective of the monthly time series analysis described in the second paragraph on page 8?

Thank you for pointing out that we were not sufficiently clear in describing the modelling methods. We have now edited the methods section throughout in order to make our methods clearer.

In summary, we fitted time dependent models stratified by age group predicting the monthly absolute inpatient mortality and admission rates. The results of this analysis are presented in the main paper. We fit a further time dependent model stratified by age group predicting absolute inpatient mortality which is adjusted for % of relevant population who were male, the population median age and mean Charlson score. This is presented in supplementary material 6.

3. In the discussion, the authors detail what changes they'd expect from the guidelines (reduced admission rates following CT imaging, reduced mortality following early diagnosis and transfer to an

appropriate level of care). It would be nice to introduce these expected directions of change earlier in the paper.

We have added the 2nd sentence of the 3rd paragraph of the background section (page 4) in order to highlight the expected directions of change.

4. Consider removing the 65+ and 0-15 analysis from this paper. It seems that the treatment methodology for 65+ is likely different from younger people due to greater levels of comorbidity and frailty. They may be less sensitive to changes in guideline care due to already high levels of CT use and transfer. Likewise for the youngest members on the sample. Those 16-64 likely represent a population at greater probability of being affected by changes in guidelines for care of TBI.

We believe that the 65+ and 0-15 analysis is important and should be included for the following reasons:

1) Time series analysis using complete national data for these outcomes has not been previously completed for any of these age groups and therefore all the analysis that we present adds to the literature.

2) As shown in supplementary material 1, different guideline iterations contained specific imaging recommendations for the 0-15, 16-64 and 65+ age groups and therefore it is important to assess whether these age group specific recommendations were associated with changes in the outcomes we are assessing.

3) As outlined in the penultimate paragraph of the comparison to previous literature section (page 19 and 20) and implications section, the apparent failure of the 2nd NICE guideline to reduce the death rate in those over 65 could be due to several factors such as unequal access to specialist treatment or that specialist treatment is less effective in this group. Either way, the observed increase in TBI death and admissions in this group represents a significant health service challenge that is worth highlighting.

Minor Comments/Typos:

1. Page 9, sentence “the proportion of all TBI annual admissions and deaths in patients over 65 increased from 17% and 49% in 1998 to 48% and 78% in 2016” is confusing. I believe you are saying that in 1998, admissions for TBI in those over 65 accounted for 17% of all TBI admissions which increased to 48% in 2016 and of those who died from a TBI, 49% were over 65 in 1998 while 78% were in 2016. Consider revision.

Yes, you are correct and we have changed this sentence accordingly.

2. Throughout the paper, either a separator in numbers is omitted (e.g., 100000) or both a comma and space (e.g., 100, 000) are used. I'd encourage use of a separator in order to make the numbers more easily understood and using either a space or a comma and not both.

We have revised the text to use a space throughout.

VERSION 2 – REVIEW

REVIEWER	Jacob Simmering University of Iowa, USA
REVIEW RETURNED	26-Mar-2019

GENERAL COMMENTS

The revisions have adequately addressed my concerns.

A probable typographic error occurs on page 8, in the methods: "A discontinuity in the gradient (level) or intercept (trend) of the fitted model was tested for at each time point..." I believe it should read "gradient (trend) or intercept (level)".